# Anorexia Nervosa Is Associated with a Shift to Pro-Atherogenic Low-Density Lipoprotein Subclasses

**DOI:** 10.3390/biomedicines10040895

**Published:** 2022-04-13

**Authors:** Julia T. Stadler, Sonja Lackner, Sabrina Mörkl, Nathalie Meier-Allard, Hubert Scharnagl, Alankrita Rani, Harald Mangge, Sieglinde Zelzer, Sandra J. Holasek, Gunther Marsche

**Affiliations:** 1Division of Pharmacology, Otto Loewi Research Center for Vascular Biology, Immunology and Inflammation, Medical University of Graz, Universitätsplatz 4, 8010 Graz, Austria; julia.stadler@medunigraz.at (J.T.S.); alankrita.rani@medunigraz.at (A.R.); 2Division of Immunology, Otto Loewi Research Center, Medical University of Graz, Heinrichstraße 31a, 8010 Graz, Austria; sonja.lackner@medunigraz.at (S.L.); nathalie.allard@medunigraz.at (N.M.-A.); 3Department of Psychiatry and Psychotherapeutic Medicine, Medical University of Graz, Auenbruggerplatz 31, 8036 Graz, Austria; sabrina.moerkl@medunigraz.at; 4Clinical Institute of Medical and Chemical Laboratory Diagnostics, Medical University of Graz, Auenbruggerplatz 15, 8036 Graz, Austria; hubert.scharnagl@medunigraz.at (H.S.); harald.mangge@medunigraz.at (H.M.); sieglinde.zelzer@medunigraz.at (S.Z.)

**Keywords:** anorexia nervosa, lipoprotein subclasses, VLDL, small LDL particles, HDL function CETP, LCAT

## Abstract

Anorexia nervosa (AN) is a severe eating disorder affecting primarily female adolescents and younger adults. The energy deprivation associated with AN has been shown to alter lipoprotein metabolism, which may affect cardiovascular risk. However, the mechanisms leading to alterations in the composition, structure, and function of lipoproteins in AN patients are not well-understood yet. Here, we investigated the lipid abnormalities associated with AN, particularly changes in the distribution, composition, metabolism, and function of lipoprotein subclasses. In this exploratory study, we analyzed serum samples of 18 women diagnosed with AN (BMI < 17.5 kg/m^2^) and 24 normal-weight women (BMI from 18.5–24.9 kg/m^2^). Using the Quantimetrix Lipoprint^®^ system, we determined low-density lipoprotein (LDL) subclass distribution, including quantitative measurements of very low-density lipoprotein (VLDL), intermediate density lipoprotein (IDL) and high-density lipoprotein (HDL) subclass distribution. We quantified the most abundant apolipoproteins of HDL and assessed lecithin-cholesterol acyltransferase (LCAT) and cholesteryl-ester transfer protein (CETP) activities. In addition, anti-oxidative capacity of apoB-depleted serum and functional metrics of HDL, including cholesterol efflux capacity and paraoxonase activity were assessed. The atherogenic lipoprotein subclasses VLDL and small LDL particles were increased in AN. Levels of VLDL correlated significantly with CETP activity (r_s_ = 0.432, *p* = 0.005). AN was accompanied by changes in the content of HDL-associated apolipoproteins involved in triglyceride catabolism, such as apolipoprotein C-II (+24%) and apoA-II (−27%), whereas HDL-associated cholesterol, phospholipids, and triglycerides were not altered. Moreover, AN did not affect HDL subclass distribution, cholesterol efflux capacity, and paraoxonase activity. We observed a shift to more atherogenic lipoprotein subclasses in AN patients, whereas HDL functionality and subclass distribution were not altered. This finding underpins potential detrimental effects of AN on lipid metabolism and the cardiovascular system by increasing atherosclerotic risk factors.

## 1. Introduction

Anorexia nervosa (AN) is a life-threatening psychiatric disorder with a high mortality rate. Its onset is often in adolescence. AN is characterized by low body mass index (BMI) and often accompanied by a wrong perception of the body and fear of gaining weight [1,2]. The reduced energy availability often leads to extensive metabolic and endocrine disturbances and alterations in lipid metabolism; however, these disturbances are still poorly understood [3].

In patients with severe AN, hepatic complications often occur, characterized by elevated levels of aspartate aminotransferase (AST) and alanine aminotransferase (ALT) [4,5]. Despite low intake of saturated fat and cholesterol-rich foods, some studies have reported that total cholesterol levels are elevated in AN patients [3,6]. In a recent meta-analysis, acutely ill AN patients were shown to have elevated levels of high-density lipoproteins (HDL), low-density lipoproteins (LDL), and triglycerides [3]. Several case reports showed cardiovascular complications in AN patients, especially in the context of refeeding that may reflect lipid fluctuations [7]. Raised total cholesterol is a strong risk factor for cardiovascular disease [8]; therefore, a better understanding of AN-associated changes in lipoprotein metabolism is urgently needed.

Lipoproteins are very heterogeneous, and several different subclasses exist, which are classified by their size and density. In the context of cardiovascular risk assessment, the lipoprotein class-specific concentrations of cholesterol and triglycerides are of interest beyond their total concentrations within the plasma. LDL cholesterol is one of the most important risk factors for cardiovascular disease and remains the main target of current cardiovascular risk reduction strategies [8,9,10]. Nevertheless, many people with LDL-cholesterol levels in the normal range develop cardiovascular disease, and it has long been recognized that certain subfractions of LDL have increased atherogenic potential, which may explain this observation [11,12].

Various methods are used for identifying the subclass distribution of lipoproteins, e.g., native gradient gel electrophoresis, density gradient ultracentrifugation, nuclear magnetic resonance (NMR), or tube gel electrophoresis with the Quantimetrix Lipoprint System^®^ [13].

Depending on their maturation state, lipoprotein subclasses contain a variety of proteins and lipids and, therefore, also execute distinct functions. In recent years, especially the small and dense LDL particles have gained attention because of their strong association with coronary heart disease [14,15]. Small dense LDL particles are more susceptible to oxidation, and their size facilitates penetration into the arterial wall, serving as a source of lipids for plaque formation [16,17]. However, other lipoproteins also rich in triglycerides, such as VLDL, have been shown to be associated with adverse outcomes and are elevated in inflammatory conditions such as obesity [18,19]. In contrast, HDL particles are considered as anti-atherogenic, due to their multiple protective properties such as anti-oxidative [20], anti-inflammatory [21], and anti-thrombotic effects [22], besides their important function in promoting reverse cholesterol transport [23]. HDL particles are very heterogeneous in size and structure, and multiple different proteins, lipids, and enzymes are associated with HDL.

In this cross-sectional study, we sought to elucidate anorexia nervosa-associated changes in lipoprotein metabolism by assessing (i) subclass distribution of LDL particles, (ii) composition and function of high-density lipoproteins, and (iii) enzyme activities involved in lipoprotein metabolism.

## 2. Materials and Methods

### 2.1. Recruitment and Group Characteristics

Patients diagnosed with AN were recruited in Graz, Austria. AN patients were included based on F50.0 International Classification of Diseases criteria for AN [24] with a body mass index < 17.5 kg/m^2^. A total of 42 study participants were enrolled: 18 women diagnosed with AN and 24 normal-weight control women, all of them aged 18–30 years. The study population was a subgroup of a larger cross-sectional study (5 groups of different energy status, *n* = 107) [25,26,27,28], was conducted according to the Declaration of Helsinki, and was approved by the ethics committee of the Medical University of Graz (MUG-26-383ex13/14). All participants gave their written informed consent. The study population was enrolled according to the following inclusion criteria: female, aged between 18 and 40 years. Only metabolically healthy women that cleared the following exclusion criteria were included: acute or chronic illness or infection, alcohol or drug abuse, statin medication, severe cognitive deficits, history of digestive tract disease, history of gastrointestinal surgery, treatment with antibiotics and use of prebiotics or probiotics within the past two months, pregnancy, or breastfeeding.

#### 2.1.1. Anthropometry and Body Fat Measurement

Standard anthropometric parameters such as body height, body weight, and circumferences of waist and hip were measured. The body mass index (BMI) was calculated according to the formula BMI = body weight [kg]/body height [m]^2^ [29].

The body fat of the participants was determined by a standardized ultrasound technique [30]. More specifically, we determined the sum of the participants’ subcutaneous adipose tissue (SAT) thicknesses at eight defined body sites (DINCL) which has been shown to be an accurate method for precisely assessing even extremely low amounts of body fat as expected for the AN group [31].

#### 2.1.2. Clinical Laboratory Parameters and Adipokines

Blood draws were conducted in overnight fasted participants. The clinical blood parameters for liver function such as alkaline phosphatase (AP), gamma-glutamyl transpeptidase (GGT), cholinesterase (CHE), alanine-transaminase (ALT), and aspartate aminotransferase (AST) as well as glycated hemoglobin (Hba1c) and C-reactive protein (CRP) were determined by clinically established standards procedures on a Cobas 6000 chemical routine analyzer (Roche Diagnostics, Mannheim, Germany). Total cholesterol, triglycerides, and HDL-cholesterol were measured by enzymatic photometric transmission measurement (Roche Diagnostics, Mannheim, Germany), and the concentrations of LDL-cholesterol were calculated by the Friedewald’s formula. The adipokines adiponectin and leptin and soluble leptin receptors (sOB-R) were determined by performing specific enzyme-linked immunosorbent assays (all BioVendor, Brno, Czech Republic).

### 2.2. Lipoprotein Subclass Analyses with Lipoprint System

Analyses of LDL as well as HDL subclass distribution were performed using the Lipoprint System^®^ (Quantimetrix Corp., Redondo Beach, CA, USA) according to the manufacturer’s instructions. The Lipoprint system separates lipoprotein subfractions from human serum based on their size by proportional binding of a lipophilic dye to the relative amount of cholesterol in the lipoproteins [32]. Briefly, 25 μL of serum, for the normal-weight controls and anorexic patients, was added to polyacrylamide gel tubes together with 300 μL loading gel solution for the HDL kit and 200 μL for the LDL kit. The loading gel contained a lipophilic dye to stain cholesterol of lipoproteins, and the gel tubes were photopolymerized at room temperature for 30 min.

Electrophoresis with tubes containing sera and manufacturer’s quality controls took 50 min for the HDL kit and 1 h for the LDL kit (3 mA/gel tube). After electrophoresis, the gel tubes were allowed to rest for 1 h. The obtained subfraction bands were scanned using an ArtixScan M2 digital scanner (Microtek International Inc., Redondo Beach, CA, USA) and analyzed using Lipoprint^®^ software.

### 2.3. HDL-Associated Proteins and Lipids

The HDL-associated apolipoproteins apoA-I, apoA-II, apoC-II, apoC-III, and apoE were assessed by immunoturbidimetry as described before [33]. Levels of serum amyloid A (SAA) were determined using a commercially available kit (Invitrogen, Carlsbad, CA, USA) according to the manufacturer’s instructions.

Lipids associated with HDL, such as free cholesterol, cholesteryl-ester, triglycerides, and phospholipids were assessed using enzymatic methods [34]. Analyses were performed on an Olympus AU680 analyzer (Beckman Coulter, Brea, CA, USA).

### 2.4. ApoB-Depleted Serum

For the analyses of HDL composition and function, apoB-depleted serum was used. A 40 µL amount of polyethylene glycol (Sigma Aldrich, Darmstadt, Germany) (20% in 200 mmol/L glycine buffer) was added to 100 µL serum, mixed gently, and then incubated for 20 min at room temperature. After centrifugation at 10,000 rpm for 30 min at 4 °C, the supernatant was collected. Samples were stored at −70 °C.

### 2.5. Anti-Oxidative Capacity of ApoB-Depleted Serum

As previously described [35], the anti-oxidative capacity of apoB-depleted serum was assessed with a fluorometric assay using the fluorescent dye dihydrorhodamine.

### 2.6. Arylesterase Activity of Paraoxonase 1 (PON1)

A photometric assay was used to determine the arylesterase activity of PON1 in apoB-depleted serum, as described elsewhere [36].

### 2.7. Cholesterol Efflux Capacity of ApoB-Depleted Serum

The cholesterol efflux capacity of apoB-depleted serum was assessed, as described elsewhere [18,23]. Briefly, J774.2 cells (Sigma Aldrich, Darmstadt, Germany) were cultured in DMEM medium (Life Technologies, Carlsbad, CA, USA) containing 10% FBS and 1% penicillin/streptomycin. A total of 300,000 cells/well were seeded in 48-well plates and cultured for 24 h. Then, cells were labeled with 0.5 µCi/mL radiolabeled [^3^H]-cholesterol (Hartmann Analytic, Braunschweig, Germany) in DMEM media containing 2% FBS, 1% penicillin/streptomycin, and 8(4-chlorophenylthio)-cyclic adenosine monophosphate (0.3 mM) (Sigma-Aldrich, Darmstadt, Germany) overnight. Next, cells were rinsed twice with DMEM media and equilibrated with serum-free DMEM containing 2% bovine serum albumin (Sigma-Aldrich, Darmstadt, Germany) for 2 h. After rinsing, sample containing 2.8% apoB-depleted serum was added and incubated for 3 h. Cholesterol efflux capacity was expressed as radioactivity in the media relative to total radioactivity of media and lysed cells. All steps were conducted in the presence of 2 µg/mL of acyl-coenzyme A cholesterol acyltransferase inhibitor (Sandoz 58-035, Sigma-Aldrich, Darmstadt, Germany).

### 2.8. LCAT Activity

LCAT activity of serum was determined using a commercially available kit (Merck, Darmstadt, Germany) according to the manufacturer’s instructions. Briefly, serum samples were incubated with the LCAT substrate for 4 h at 37 °C. The substrate emits fluorescence at 470 nm. Hydrolysis of the substrate by LCAT releases a monomer that emits fluorescence at 390 nm. LCAT activity is measured over time and expressed as the change in emission intensity at 470/390 nm.

### 2.9. CETP Activity

CETP activity of serum was assessed by a commercially available kit (Abcam, Cambridge Science Park, Cambridge, UK) according to the manufacturer’s instructions. The assay uses a donor molecule containing a fluorescent self-quenched neutral lipid that is transferred to an acceptor molecule in the presence of CETP. The CETP-mediated transfer of the fluorescent lipid to the acceptor molecule results in an increase in fluorescence intensity (excitation: 465 nm; emission: 535 nm).

### 2.10. Statistical Analyses

For statistical analysis of this explorative study, the software SPSS Statistics version 26.0 and GraphPad Prism version 6 were used. The Shapiro–Wilk test was used to assess normal distribution of the data. Depending on the normal distribution of the data, *t*-test or Mann–Whitney U test was used to analyze differences between the two groups. For multiple testing, *p*-values were adjusted according to Benjamini–Hochberg to decrease the false discovery rate. Data are presented as median and interquartile range. Correlations were determined using Spearman’s correlation coefficient r_s_. Our study (*n* = 24 vs. *n* = 18) provided a power greater than 90% to detect a 10% difference in VLDL-cholesterol, based on our hypothesis that we would observe an effect of low BMI on atherogenic lipoproteins [18].

## 3. Results

The clinical characteristics of the study cohort are shown in Table 1. Patients with AN had a median BMI of 15.5 kg/m^2^ and a D_INCL_ (thickness of subcutaneous adipose tissue at eight measured body sites) of 30.2 mm. Normal-weight controls did not differ in levels of plasma lipids from the AN group. Although levels of liver markers were still in a normal and healthy range, patients with AN showed differences in GGT, CHE, ALT, and AST levels. The AN patients had lower levels of CRP, highlighting that they did not suffer from inflammatory conditions despite a low BMI. As expected, serum levels of the adipokine leptin were lower in the AN group; however, serum concentrations of the soluble leptin receptor (sOB-R) and adiponectin were increased.

### 3.1. Anorexia Nervosa Is Associated with Changes in Subclass Distribution of Triglyceride-Rich Lipoproteins

Studies have shown that AN is often accompanied with elevated levels of plasma lipids; however, nothing is known about the effect of low BMI on the distribution or composition of lipoproteins. Therefore, the subclass distribution of low-density lipoproteins was assessed using the Lipoprint system^®^ (Figure 1). A representative lipoprotein distribution chart of one normal-weight control and one AN patient each is shown (Figure 1H). Of particular interest, the analyses revealed a profound increase in VLDL in AN patients, while levels of the intermediate density lipoprotein (IDL) subclasses IDL-B and IDL-A were decreased, suggesting disturbances in VLDL hydrolyzation (Figure 1A–D).

Interestingly, we further observed changes in the subclass distribution of LDL (Figure 1E–G). In particular, the large LDL subclass, which is represented by subclass LDL1, was decreased in AN patients and shifted to the medium (LDL2) and small LDL subclasses (LDL3–LDL7) (*p* = 0.039), indicating a shift towards lipoprotein subclasses with increased atherogenic potential.

### 3.2. Anorexia Nervosa Was Associated with Changes in HDL Composition, but Not with Significant Alterations in HDL Subclass Distribution or HDL Function

An increased BMI is associated with changes in HDL metabolism [18]. Therefore, we were also intrigued by potential effects of low BMI on HDL composition, subclass distribution, and function.

HDL particles are very heterogeneous and differ in their density, size, and composition, which in turn determines the functionality of the particles. Using the Lipoprint system, 10 different HDL subgroups were identified, of which three subclasses, termed large, medium, and small HDL, were categorized. We observed no significant changes in the distribution of HDL subclasses when comparing normal-weight controls and anorexic patients (Figure 2).

HDL particles exert many protective properties, including anti-oxidative capacity. In our study, we observed no changes in apoB-depleted serum of AN women in the inhibition of the radical-induced oxidation of the fluorescent dye dihydrorhodamine, when comparing with normal-weight women (Figure 3A). Additionally, neither the activity of HDL-associated enzyme paraoxonase 1 (PON1) (Figure 3B) nor the cholesterol efflux capacity (Figure 3C) was altered in women with AN.

Despite the important role of HDL lipids and related proteins in HDL functionality, it is currently unknown whether AN leads to changes in HDL composition. Therefore, we assessed how a low BMI affects the constitution of important apolipoproteins in AN. Remarkably, we observed a decrease in the second most abundant apolipoprotein, apoA-II (Figure 4B), whereas the levels of the major protein apoA-I were not different (Figure 4A). The apoCs are involved in the regulation of lipoprotein lipase (LPL), an enzyme that catalyzes the degradation of triglycerides from circulating VLDL and chylomicrons [37]. Of particular interest, levels of apoC-II, an essential cofactor of LPL activation [38,39], were increased in AN patients compared to normal-weight controls (Figure 4C). However, levels of HDL-associated apoC-III, which is by contrast an inactivator of LPL [40], were not altered (Figure 4D). Interestingly, levels of the pro-inflammatory protein serum amyloid A (SAA) were decreased in the AN study group (Figure 4F) and correlated significantly with CRP (r_s_ = 0.78, *p* < 0.001). No differences were observed for levels of apoE (Figure 4E).

Due to the AN-associated alterations in HDL apolipoprotein composition, we next were interested in whether a low BMI also affects HDL-associated lipids. In apoB-depleted serum, levels of free cholesterol, cholesteryl-ester, phospholipids, and triglycerides were assessed. However, HDL lipid composition did not differ between AN women and normal-weight women (Figure 5A–D).

### 3.3. Effect of Anorexia Nervosa on Enzymes Involved in Lipoprotein Metabolism

We assessed the activities of enzymes that are important in lipoprotein metabolism. The cholesteryl-ester transfer protein (CETP) transfers cholesteryl-esters from HDL to VLDL in exchange for triglycerides. The activity of this lipid transfer protein did not differ significantly between normal-weight controls and AN patients (Figure 6A), but we observed a significant correlation between CETP activity and VLDL levels (r_s_ = 0.432, *p* = 0.005). Another crucial enzyme in HDL metabolism is the lecithin-cholesteryl acyltransferase (LCAT). This enzyme plays a key role in maturation of HDL by esterifying free cholesterol on the surface of HDL, leading to the formation of the hydrophobic inner cholesteryl-ester core of HDL and increasing the size of the particles. In addition, this step is also important for the capacity of HDL to perform reverse cholesterol transport [41]. We assessed LCAT activity in the AN group compared with that in normal-weight women. After correction according to Benjamini–Hochberg, no significant difference was observed (*p* = 0.133) (Figure 6B).

## 4. Discussion

In this study, we demonstrated that AN is associated with markedly altered VLDL/LDL subclasses, even though we did not observe significant changes in routinely determined plasma lipids. Multiple mechanisms may affect the lipid metabolism in AN patients, including dietary, absorptive, and genetic factors [3]. It is likely that the reduced energy availability in AN contributes to the disturbances in lipid metabolism, such as hormonal alterations leading to altered lipolysis in peripheral tissues. Furthermore, the lack of energy may result in reduced endogenous cholesterol synthesis, which affects LDL removal and steroid hormone synthesis [3,6,42]. To our knowledge, this is the first study to show an AN-associated increase in VLDL and a shift from the large LDL subclass to the more pro-atherogenic small LDL subclass. Elevated VLDL cholesterol levels have been shown to be directly associated with an increased risk of heart attack [43].

The lipoprotein subclass analysis further revealed that in addition to changes in VLDL and IDL subclasses, the major LDL subclass distribution also was altered in AN. Specifically, we observed a shift from the large to small LDL subfractions. Of particular interest, elevated levels of small, dense LDL particles are associated with increased cardiovascular risk [10,44]. Small, dense LDL particles are more susceptible to oxidation and are an important source of lipids contributing to atherosclerotic plaque formation in the arterial wall [17]. Cardiac abnormalities have been identified with echocardiography in AN, which is considered to aid the effectiveness and safety of nutritional rehabilitation in addition to the serum lipid profile [45]. Given that mortality rates are five-fold higher in AN when compared with the general population [46], a shift to more atherogenic lipoprotein subclasses may also indicate a higher cardiovascular risk.

The causes of AN are unknown. Other than being female, few risk factors for the disease have been identified. Some genetic and social factors play a role in the development of AN [47,48,49]. The desire to be slim is widespread in Western society, and even before puberty, children are aware of this attitude, and more than half of preadolescent girls diet or adopt other measures to control their weight. However, only a small percentage of these girls develop AN. Other factors, such as psychological vulnerability, probably predispose certain individuals to the development of AN. Many affected individuals belong to the middle or upper socioeconomic classes, are meticulous and compulsive, and have very high standards of achievement and success.

Interestingly, elevated serum levels of VLDL are also observed in obesity when compared with normal-weight individuals, but in obesity also, HDL subclass distribution and function are affected [18]. The size of HDL particles affects their functionality, and the small HDLs have been shown to be the most protective [50]. However, we observed no changes in HDL subclass distribution in AN patients. Consistent with this observation, we also were unable to detect changes in parameters of HDL function, such as cholesterol efflux capacity or PON1 activity. Furthermore, anti-oxidative capacity of apoB-depleted serum was not affected by AN.

Of particular interest, when assessing HDL apolipoprotein composition, we observed a decrease in HDL-associated apoA-II. This apolipoprotein is the second most abundant protein on HDL, is mainly synthesized by the liver, and has been suggested to be important in HDL particle maturation [51]. Moreover, experiments with transgenic mice overexpressing human apoA-II have shown a decreased ratio of esterified cholesterol/total cholesterol, due to decreased LCAT activity [52,53]. Overexpression of apoA-II is linked to hypertriglyceridemia. It is thought that apoA-II decreases lipoprotein lipase activity and subsequent VLDL hydrolysis [52]. On the other hand, we observed increased HDL-associated apoC-II levels. ApoC-II activates lipoprotein lipase, whereas apoC-III is an inhibitor of lipoprotein lipase, but HDL-associated levels of apoC-III were not altered [54]. Similarly, an increase in apoC-II was also shown by another study, besides elevated serum levels of apoA-I, apoB, apoC-III, and apoE [55]. Interestingly, the levels of the acute phase protein serum amyloid A were lower in AN patients compared to normal-weight women, highlighting that our AN patients showed no sign of inflammation. In line with this observation, serum levels of C-reactive protein were also lower in AN and correlated strongly with serum amyloid A.

Importantly, our study further revealed AN-associated changes in the activity of enzymes that are important in lipoprotein metabolism. It has been previously reported that CETP activity is elevated in AN patients [55,56]. This increase was linked to accelerated cholesterol turnover and suggested as an adaption to the low cholesterol intake usually observed in AN patients [56]. However, in our study, CETP activity was not altered in AN patients but correlated significantly with plasma triglycerides (r_s_ = 0.419, *p* = 0.006) and VLDL levels (r_s_ = 0.432, *p* = 0.005), in line with the CETP-mediated transfer of cholesteryl-esters from HDL to VLDL in exchange for triglycerides. LCAT is crucial for the maturation of HDL particles and has been shown to be involved in reverse cholesterol transport, which is an important anti-atherogenic function of HDL [57]. We observed that LCAT activity showed a non-significant trend toward increased levels in AN patients (*p* = 0.133). Future, larger studies are needed to demonstrate whether AN is associated with increased LCAT activity or not.

We acknowledge some further limitations to this study. We used the Lipoprint system^®^, an FDA approved and very gentle method to separate lipoprotein subclasses according to their size. Since separation methods are based on different principles, the estimated number of subclasses depends on the method and may be different for other methods. No heparinized serum was available from this study cohort; therefore, measurement of lipoprotein lipase or hepatic lipase activities was not possible. The number of patients in this explorative study was limited; therefore, further studies in larger cohorts are needed to confirm our results and to draw firm conclusions.

## 5. Conclusions

In conclusion, we demonstrated that AN is associated with an increase in serum levels of VLDL and a shift toward small pro-atherogenic LDL subclasses, whereas no differences in the distribution and function of HDL subclasses were found. Taken together, our results suggest a detrimental shift of lipoprotein subclasses that may contribute to increased cardiovascular risk in AN patients.

## Figures and Tables

**Figure 1 biomedicines-10-00895-f001:**
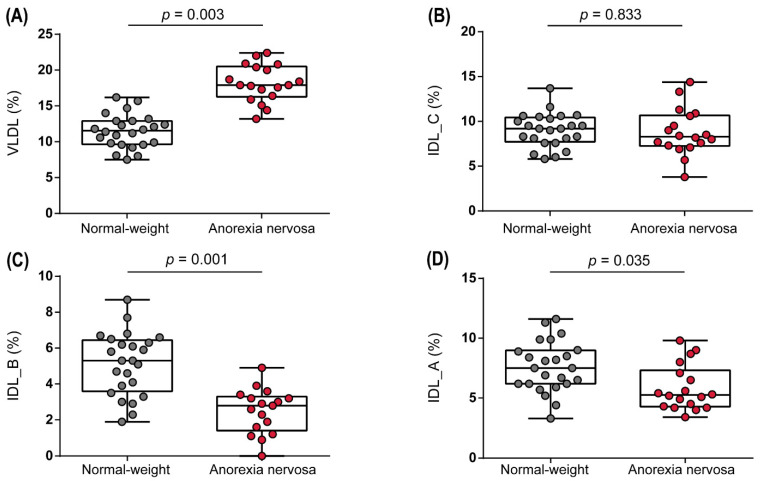
Subclass distribution of low-density lipoproteins. Using the Lipoprint system^®^, lipoproteins were separated into VLDL (**A**) and the intermediate-density lipoprotein subclasses IDL-C (**B**), IDL-B (**C**) and IDL-A (**D**). Distribution of LDL subclasses was assessed (**E**–**G**). The percentage of each subclass was calculated as the proportion of total plasma cholesterol measured before the analyses. A representative Lipoprint result chart of one normal-weight and one AN woman is shown (**H**). Depending on the normal distribution of the data, *t*-test (**A**–**F**) or Mann–Whitney U test (**G**) was used to analyze differences between the two groups. Individual data are presented on top of boxplots showing median, interquartile range, and minimum and maximum values. Adjusted *p*-values according to Benjamini–Hochberg are displayed.

**Figure 2 biomedicines-10-00895-f002:**
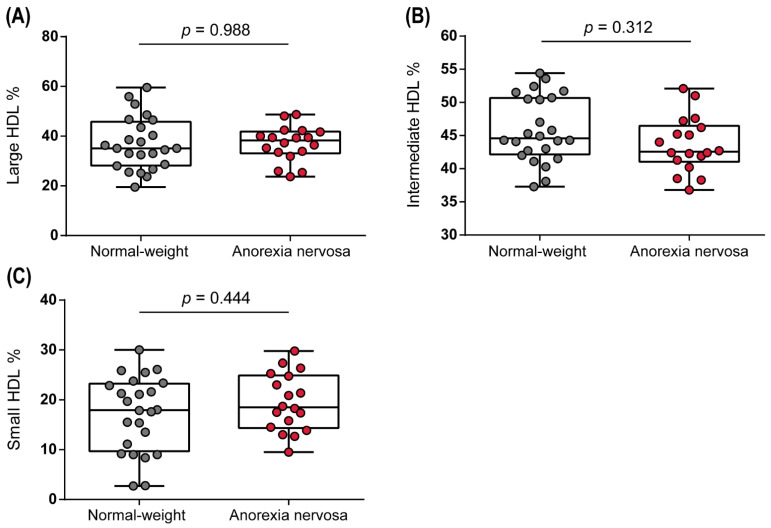
HDL subclass distribution of normal-weight and anorexia nervosa women. Subfractions of HDL were identified using the Lipoprint system. Subclasses were categorized into large (**A**), intermediate (**B**), and small HDL (**C**). Depending on the normal distribution of the data, *t*-test or Mann–Whitney U test was used to analyze differences between the two groups. Individual data are presented on top of boxplots, displaying median, interquartile range, as well as minimum and maximum values. Adjusted *p*-values according to Benjamini–Hochberg are displayed.

**Figure 3 biomedicines-10-00895-f003:**
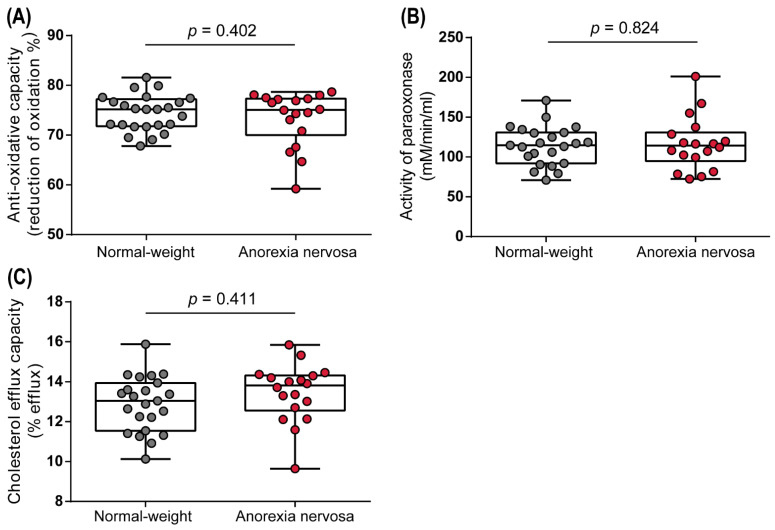
Parameters of HDL function in normal-weight and anorexia nervosa women. The anti-oxidative capacity (**A**) as well as arylesterase activity of PON1 (**B**) were assessed in apoB-depleted serum. Cholesterol efflux capacity was determined using a cell-based assay (**C**). Depending on the normal distribution of the data, *t*-test or Mann–Whitney U test was used to analyze differences between the two groups. Individual data are presented on top of boxplots, displaying median, interquartile range, as well as minimum and maximum values. Adjusted *p*-values according to Benjamini–Hochberg are displayed.

**Figure 4 biomedicines-10-00895-f004:**
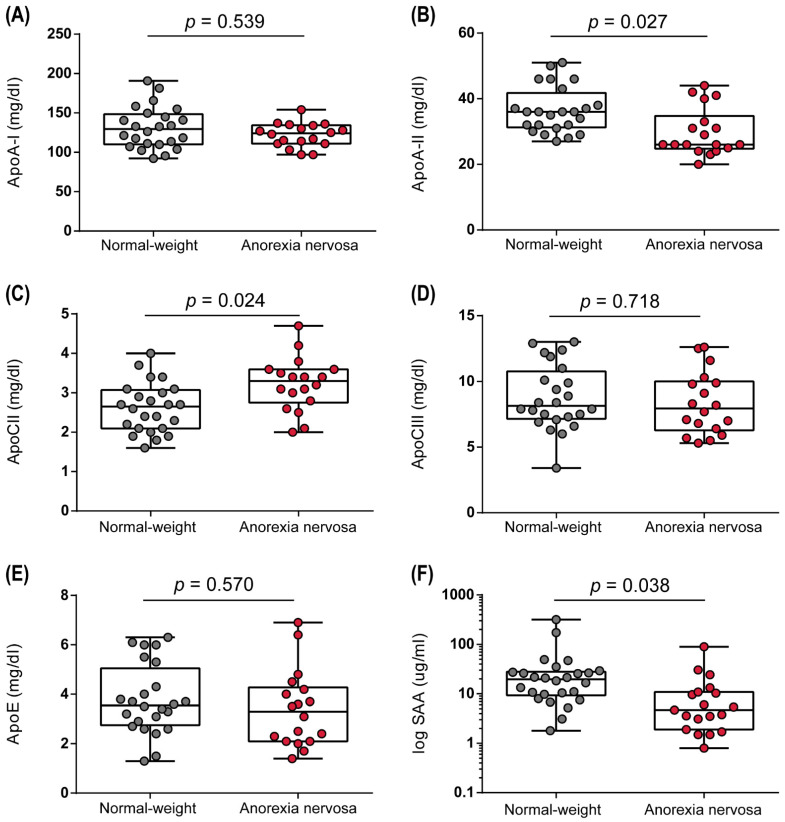
Apolipoprotein composition of HDL in normal-weight and anorexia nervosa women. HDL-associated proteins including apoA-I (**A**), apoA-II (**B**), apoC-II (**C**), apoC-III (**D**), apoE (**E**), and SAA (**F**) were determined in apoB-depleted serum. Depending on the normal distribution of the data, *t*-test or Mann–Whitney U test was used to analyze differences between the two groups. Individual data are presented on top of boxplots, displaying median, interquartile range, as well as minimum and maximum values. Adjusted *p*-values according to Benjamini–Hochberg are displayed.

**Figure 5 biomedicines-10-00895-f005:**
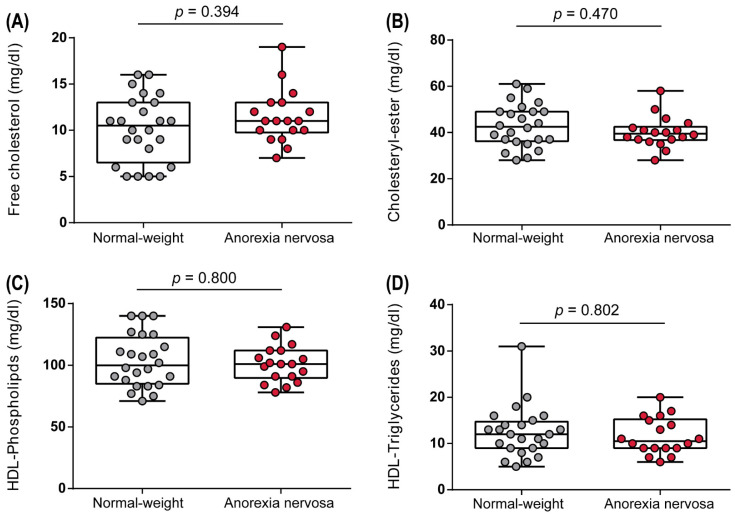
Composition of HDL-associated lipids. Levels of free cholesterol (**A**), cholesteryl-ester (**B**), phospholipids (**C**), and triglycerides (**D**) were determined in the study cohort. Depending on the normal distribution of the data, *t*-test or Mann–Whitney U test was used to analyze differences between the two groups. Individual data are presented on top of boxplots, displaying median, interquartile range, as well as minimum and maximum values. Adjusted *p*-values according to Benjamini–Hochberg are displayed.

**Figure 6 biomedicines-10-00895-f006:**
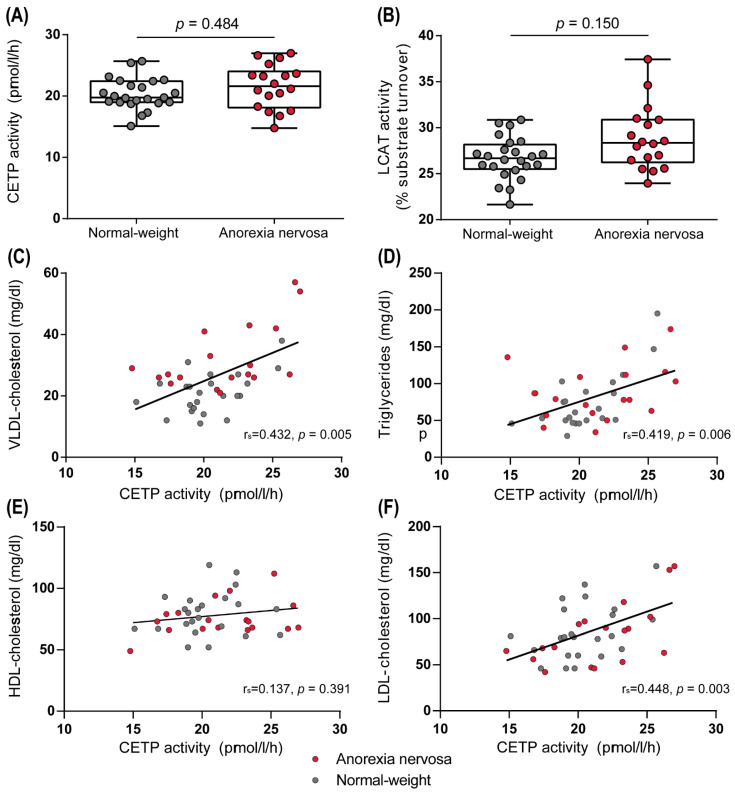
Activities of important enzymes in HDL metabolism are affected in anorexia nervosa. The activities of cholesteryl-ester transfer protein (CETP) (**A**) and lecithin-cholesteryl acyltransferase (LCAT) (**B**) were assessed in serum samples from the study population. Correlation analysis between CETP activity and plasma lipid levels was performed (**C**–**F**). Depending on the normal distribution of the data, *t*-test or Mann–Whitney U test was used to analyze differences between the two groups. Individual data are presented on top of boxplots, showing median, interquartile range, as well as minimum and maximum values. Adjusted *p*-values according to Benjamini–Hochberg are displayed.

**Table 1 biomedicines-10-00895-t001:** Clinical characteristics of the study cohort.

Study Characteristics	Anorexia Nervosa (*n* = 18)	Normal-Weight (*n* = 24)	*p*-Value
Age (years)	22 (19–25)	24 (22–26)	0.082
BMI (kg/m^2^)	15.5 (14.3–16.2)	21.8 (20.2–23.5)	<0.001
D_INCL_ (mm)	30.2 (9.9–46.0)	83.6 (66.1–99.1)	<0.001
HDL-cholesterol (mg/dL)	73.0 (67.0–81.5)	79.5 (67.0–89.3)	0.461
LDL-cholesterol (mg/dL)	78.0 (55.3–98.3)	80.0 (60.8–108.5)	0.797
Triglycerides (mg/dL)	78.5 (59.3–113.0)	63.5 (50.0–98.8)	0.199
Total cholesterol (mg/dL)	169.0 (145.3–202.0)	167.0 (151.3–207.0)	0.692
CRP (mg/L)	0.6 (0.6–1.6)	1.4 (0.6–2.5)	0.013
Hba1c	31.0 (29.5–34.3)	31.0 (29.3–33.0)	0.547
Leptin (ng/mL)	1.6 (1.0–3.8)	10.6 (7.4–14.4)	<0.001
sOB-R (ng/mL)	30.8 (24.4–39.3)	19.3 (16.4–21.8)	<0.001
Adiponectin (μg/mL)	16.5 (12.7–20.8)	11.5 (9.3–16.7)	0.019
AP (U/L)	52.5 (42.8–57.3)	50.0 (44.0–55.8)	0.865
GGT (U/L)	19.5 (11.5–24.3)	12.5 (11.0–16.8)	0.023
CHE (U/L)	6727 (5900–7201)	7229 (6676–7901)	0.038
ALT (U/L)	23.0 (15.0–34.0)	14.0 (12.0–19.8)	0.003
AST (U/L)	21.0 (20.0–23.3)	20.0 (17.3–24.3)	0.151

Clinical characteristics of study participants. Data are shown as median (Q1–Q3). Differences between normal-weight women and anorexia nervosa patients were assessed using Mann–Whitney U-test. *n*, number of study participants; BMI, body mass index; D_INCL_, thickness of subcutaneous adipose tissue at eight measured body sites; HDL, high-density lipoprotein; LDL, low-density lipoprotein; CRP, C-reactive protein; HbA1c, glycated hemoglobin A1c; sOB-R, soluble leptin receptor; AP, alkaline phosphatase, GGT, gamma-glutamyl transpeptidase; CHE, cholinesterase; ALT, alanine transaminase; AST, aspartate aminotransferase.

## Data Availability

Data is contained within the article.

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
