# Peer review of "Anorexia Nervosa Is Associated with a Shift to Pro-Atherogenic Low-Density Lipoprotein Subclasses"

_biomedicines, 2022, doi:10.3390/biomedicines10040895_

Round 1
Reviewer 1 Report
It is a very interesting research, well described and commented.
The topic is well addressed, the methodology is adequate and the presentation is clear and follows a sound reasoning.
The only weakness is something which happens anytime one does detailed tests: the number of cases. They might be asked if they have calculated the representativeness of the sample size.
Author Response
Reviewer 1:
General Comment:
It is a very interesting research, well described and commented.
The topic is well addressed, the methodology is adequate and the presentation is clear and follows a sound reasoning.
The only weakness is something which happens anytime one does detailed tests: the number of cases. They might be asked if they have calculated the representativeness of the sample size.
-------------------------------------------------------------------------------------------------------------------------------------------
We are grateful for this encouraging feedback.
We agree with the reviewer that the sample size is small. However, we must note that this is an exploratory study, and the data in the present manuscript are additional analyses of a previously published study cohort (DOI: 10.1002/eat.22801).
According to the reviewers suggestion, we performed a power analysis and added the following text in the method section :”Our study (n=24 vs n=18) provided a power greater than 90% to detect a 10% difference in VLDL-cholesterol, based on our hypothesis that we would observe an effect of low BMI on atherogenic lipoproteins as seen in our previous study (reference 8, doi: 10.3390/biomedicines9030242)”.
Reviewer 2 Report
The authors aimed to compare anorexia nervosa and normal weight women for their lipoprotein metabolism, which was modeled in terms of LDL subclass distribution, LDL composition and function and enzymatic activity related to lipoprotein metabolism.
The paper is well written and the problem to address is clearly described, however the sample size does not seem to be appropriate to achieve the paper’s objective because the number of variables involved seems to be almost as large as the number of subjects.
In addition, the problem to tackle is intrinsically multivariate, which raises the following questions that must be explicitly addressed by the authors, since correctness of their methods of data analysis and robustness of conclusions are in jeopardy:
- A multivariate problem should not be addressed by using univariate methods of data analysis, such as the Mann-Whitney (MW) test, because it does not account for the effect of correlations among the dependent variables.
- Therefore, without proper justification, it is not guaranteed that the whole statistical inference process was conducted at the 5% significance level. How have the authors controlled for inflation of type I error probability in the consecutive comparisons performed with the MW test?
- It is argued that the MW test was chosen due to non-normality of data distribution. However, deviation from normality, per se, is not a determinant reason to abandon a parametric approach to the data, because lack of normality does not prevent such approach, as long as the data are not excessively skewed, which does not seem to be the case for many variables depicted in the boxplots.
- In view of the multivariate nature of their data, why did the author not try to address these comparisons using MANOVA on appropriately sized variable vectors, should the assumptions for MANOVA be valid?
In sum, I am of the opinion that the paper requires a major revision before it can be considered for publication.
Author Response
Reviewer 2:
General Comment:
The authors aimed to compare anorexia nervosa and normal weight women for their lipoprotein metabolism, which was modeled in terms of LDL subclass distribution, LDL composition and function and enzymatic activity related to lipoprotein metabolism.
The paper is well written and the problem to address is clearly described, however the sample size does not seem to be appropriate to achieve the paper’s objective because the number of variables involved seems to be almost as large as the number of subjects.
We thank the reviewer for reviewing our manuscript and providing the valuable feedback and suggestions for improvement.
We agree with the reviewer that further larger studies are needed to confirm our observations and to draw firm conclusions, as stated in the limitations section. Unfortunately, because this is an exploratory study and the data in this manuscript are additional analyses of a previously published study (DOI: 10.1002/eat.22801), we cannot increase the sample size.
In addition, the problem to tackle is intrinsically multivariate, which raises the following questions that must be explicitly addressed by the authors, since correctness of their methods of data analysis and robustness of conclusions are in jeopardy:
- A multivariate problem should not be addressed by using univariate methods of data analysis, such as the Mann-Whitney (MW) test, because it does not account for the effect of correlations among the dependent variables.
We thank the reviewer for this comment. As the reviewer pointed out in his/ her general comment, the sample size of our cohort is limited since the data in the current manuscript are additional analyses of a previous published study. To our knowledge, multivariate statistical approaches would necessitate larger sample sizes. Therefore, we do not think that the number of subjects included would be appropriate to perform multivariate analysis using MANOVA. The results of our study are explorative only and highlight descriptive associations. However, we agree with the reviewer that in-depth data analysis in further larger study cohorts are needed to draw firm conclusions. We highlight this in the revised manuscript (limitation section).
- Therefore, without proper justification, it is not guaranteed that the whole statistical inference process was conducted at the 5% significance level. How have the authors controlled for inflation of type I error probability in the consecutive comparisons performed with the MW test?
We thank the reviewer for pointing out this issue. For our multiple analysis we have now corrected all p-values according to the Benjamini-Hochberg correction to minimize the false discovery rate (type I errors).
- It is argued that the MW test was chosen due to non-normality of data distribution. However, deviation from normality, per se, is not a determinant reason to abandon a parametric approach to the data, because lack of normality does not prevent such approach, as long as the data are not excessively skewed, which does not seem to be the case for many variables depicted in the boxplots.
We are thankful for the input regarding the chosen test to compare our two groups. According to the reviewer’s suggestion, we have now used the Shapiro-Wilk test – which is more robust for smaller sample sizes - to test for a normal distribution of our data. Instead of using the Mann-Whitney test for all variables, we now used the parametric t-test to compare normally distributed variables.
- In view of the multivariate nature of their data, why did the author not try to address these comparisons using MANOVA on appropriately sized variable vectors, should the assumptions for MANOVA be valid?
We agree with the reviewer that such analyses would be useful to test the possible influence of other variables in a larger cohort. However, as already mentioned above, the sample size of our study is too small to perform in-depth data analysis and to perform multivariate analysis.
Reviewer 3 Report
The manuscript titled "Anorexia nervosa is associated with a shift to pro-atherogenic low-density lipoprotein subclasses" studied changes in lipid content in patients carrying Anorexia nervosa (AN). The authors demonstrated that patients showed lower BMI and D(INCL) but no clear differences in total neutral lipid content. Instead, the authors showed the patients had a higher content of lipoprotein subclasses, including VLDL and small LDL particles. Also, it is demonstrated that the levels of apoCII and apoAII in HDL were increased in AN patients although TAG and cholesterol levels did not show differences.
The manuscript is written with solid scientific methods and clear presentations. The introduction is enough for readers to understand the scope and the range of the results. The authors admit an explicit limitation of the work, but the data interpretation is fair.
I have a minor comment that the author can quickly address before publication.
Please add a visual demonstration about the correlation between CETP levels and plasma lipid levels.
Author Response
Reviewer 3:
General Comment:
The manuscript titled "Anorexia nervosa is associated with a shift to pro-atherogenic low-density lipoprotein subclasses" studied changes in lipid content in patients carrying Anorexia nervosa (AN). The authors demonstrated that patients showed lower BMI and D(INCL) but no clear differences in total neutral lipid content. Instead, the authors showed the patients had a higher content of lipoprotein subclasses, including VLDL and small LDL particles. Also, it is demonstrated that the levels of apoCII and apoAII in HDL were increased in AN patients although TAG and cholesterol levels did not show differences.
The manuscript is written with solid scientific methods and clear presentations. The introduction is enough for readers to understand the scope and the range of the results. The authors admit an explicit limitation of the work, but the data interpretation is fair.
---------------------------------------------------------------------------------------------------------------------------------------------
We want to thank Reviewer 3 for reviewing our manuscript and for his/her valuable comments and suggestions for further improvement.
I have a minor comment that the author can quickly address before publication.
Please add a visual demonstration about the correlation between CETP levels and plasma lipid levels.
According to the reviewer’s suggestion, we have included the visualization of the correlations between CETP and plasma lipid levels in the revised manuscript (Figure 6C-F, page 12).
Round 2
Reviewer 2 Report
I'd like the thank the authors for addressing the issues raised in my first revision. I still have the following concerns and questions, which I hope you can discuss in the text:
- All p-values shown in figures 1. to 6. regarding group comparisons are different from those in the first submission. Is this because such p-values were obtained with the t-test, instead of the Mann-Whitney test used previously? If so, the corresponding legends should clearly state which test was applied.
- What, and how large, is the "family of tests" considered in the Benjamini-Hochberg correction? In addition, which values have the authors considered as "false discovery rate" in this procedure (0.05, 0.1, 0.25?)
- Finally, the statement "We observed that LCAT activity is increased in AN patients (p = 0.048), however, the significance did not remain after p-value correction according to Benjamini-Hochberg." requires clarification, since what needs to be compared with the Benjamini-Hochberg critical values are the actual p-values obtained with the t-test (0.133) and not the corresponding p-values produced by the MW test (0.048).
Author Response
I'd like the thank the authors for addressing the issues raised in my first revision. I still have the following concerns and questions, which I hope you can discuss in the text:
We thank the reviewer for the positive feedback on our major revision and hope that we can answer the remaining questions adequately.
- All p-values shown in figures 1. to 6. regarding group comparisons are different from those in the first submission. Is this because such p-values were obtained with the t-test, instead of the Mann-Whitney test used previously? If so, the corresponding legends should clearly state which test was applied.
Thank you for pointing out this issue. With the exception of the variable small LDL, which did not pass the Shapiro-Wilk test, the comparative analyses were performed with the t-test, but this did not have a major impact on the p-values. The p-values differ from the first submitted manuscript because we adjusted them according to Benjamini-Hochberg, according to your suggestion.
We have now specified the applied test in the Figure legend of Figure 1 “Depending on the normal distribution of the data, t-test (A-F) or Mann-Whitney U test (G) was used to analyze differences between the two groups”.
- What, and how large, is the "family of tests" considered in the Benjamini-Hochberg correction? In addition, which values have the authors considered as "false discovery rate" in this procedure (0.05, 0.1, 0.25?)
As described above and at the suggestion of the reviewer, we adjusted the p-values of all tested variables according to Benjamini-Hochberg to reduce the false discovery rate (0.05).
For this correction procedure, we ordered all variables by their p-values from smallest to largest and assigned a rank to each variable. The adjusted p-values were then calculated by multiplying the uncorrected p-value with (sum of variables/rank).
To indicate this p-value correction, we have added a sentence in the figure legends. “Adjusted p-values according to Benjamini-Hochberg are displayed.”
- Finally, the statement "We observed that LCAT activity is increased in AN patients (p = 0.048), however, the significance did not remain after p-value correction according to Benjamini-Hochberg." requires clarification, since what needs to be compared with the Benjamini-Hochberg critical values are the actual p-values obtained with the t-test (0.133) and not the corresponding p-values produced by the MW test (0.048).
We thank the reviewer for this comment. We have rewritten that statement.